# The HeFQUIN Approach to Extract and to Process Composite Values with RML and SPARQL

(Demonstration Paper)

Olaf Hartig

*Linköping University, Linköping, Sweden*

### Abstract

Recent work has introduced a representation of generic types of composite values (lists and maps, in particular) as literals in RDF-based knowledge graphs, and to extend SPARQL with features related to such literals. This paper introduces an approach to capture the creation of such literals within RML mappings. We have implemented this approach in our query federation engine HeFQUIN and wish to demonstrate this feature during the workshop.

## 1. Introduction

The RDF Mapping Language (RML) is a declarative language to express mappings from heterogeneous types of data sources to RDF-based knowledge graphs (KGs). The language is supported by at least nine different systems [1] and there exists an active community[1] that strives towards official standardization of the language by the World Wide Web Consortium (W3C). As a milestone in this direction, earlier work on the language has been consolidated into a consistent collection of related specifications and a redesigned ontology as the new basis of the language [2]. As part of this effort, the community has identified the need to extend the language to capture the creation of composite structures that are represented via the so-called collections and containers vocabulary [3, Sections 5.1–5.2] that is part of the RDF standard, and a new module for this purpose has been added to RML [2, Section 5.3][4].

However, the representation of composite structures via the collections and containers vocabulary may not always be desirable because it is verbose and bloats up the storage footprint of the resulting RDF data. Moreover, extracting information from such composite structures using SPARQL queries is tricky [5], and manipulating them using SPARQL update statements even more so. Due to these issues, in recent work we have introduced an approach to represent generic types of composite values as RDF literals, and to extend SPARQL with features related to such literals [6, 7]. More specifically, the approach defines two datatypes for literals, denoted by the IRIs cdt:List and cdt:Map. While a cdt:List literal represents a list of RDF terms, a cdt:Map literal represents a map of key-value pairs in which every key is a distinct IRI or literal and every value can be an arbitrary RDF term.

In this demo paper, we now introduce a straightforward approach to capture the creation of such literals within RML mappings that are defined for JSON-based data sources.[2] We have implemented this approach in the RML-processing component of our query federation engine HeFQUIN and wish to demonstrate this feature during the workshop. The remainder of this paper presents the approach (Section 2), investigates whether the approach can already be used with existing RML processors (Section 3), and describes our implementation that we aim to demonstrate (Section 4). We assume in this paper that the reader is familiar with RDF, SPARQL, as well as RML.

## 2. Approach

Our approach is based on the fact that the datatype of cdt:List literals is defined such that any JSON array is a valid lexical form of such literals and, likewise, any JSON object is a valid lexical form of cdt:Map literals. Consequently, if an an object map of an RML triples map has an rml:reference property with a JSONPath expression that selects a JSON array (respectively, a JSON object), then the RDF term

---

*KGCW'26: 7th International Workshop on Knowledge Graph Construction, May 10, 2026, Dubrovnik, Croatia*

[1]https://www.w3.org/community/kg-construct/

[2]While the approach in its current form focuses on JSON-based data sources, the underlying idea can be extended to other types of data sources (which requires appropriate casting and serialization functionality, and is part of our future work).

```
1  { "aaa" : { "bbb" : [11,22],
2             "ccc" : { "xxx": 43,
3                       "yyy": [3,4] } }
4  }
```

**Listing 1:** Simple JSON file (`example.json`) to illustrate the approach.

```
1  PREFIX rml:  <http://w3id.org/rml/>
2  PREFIX cdt:  <http://w3id.org/awslabs/neptune/SPARQL-CDTs/>
3  PREFIX ex:   <http://example.org/>
4  _:tm  rml:logicalSource [ rml:source [ a rml:RelativePathSource;
5                                         rml:root rml:MappingDirectory;
6                                         rml:path "example.json" ] ;
7                     rml:referenceFormulation rml:JSONPath ;
8                     rml:iterator "$.aaa" ] ;
9       rml:subjectMap [ rml:termType rml:BlankNode ] ;
10      rml:predicateObjectMap [
11              rml:predicateMap [ rml:constant ex:test1 ] ;
12              rml:objectMap [ rml:reference "bbb" ; rml:datatype cdt:List ] ] ;
13      rml:predicateObjectMap [
14              rml:predicateMap [ rml:constant ex:test2 ] ;
15              rml:objectMap [ rml:reference "ccc" ; rml:datatype cdt:Map ] ] .
```

**Listing 2:** RML mapping with one triples map to illustrate the approach (written in the RDF Turtle syntax).

```
1  PREFIX cdt:  <http://w3id.org/awslabs/neptune/SPARQL-CDTs/>
2  PREFIX ex:   <http://example.org/>
3  _:b  ex:test1  "[11,22]"^^cdt:List ;
4       ex:test2  """{"xxx": 43, "yyy": [3,4]}"""^^cdt:Map .
```

**Listing 3:** RDF graph obtained by applying the RML mapping of Listing 2 to the JSON document of Listing 1.

produced by this object map can be a cdt:List literal with the JSON array as its lexical form (respectively, a cdt:Map literal with the JSON object as lexical form).

To illustrate this idea, consider the JSON document in Listing 1 and the RML triples map in Listing 2. Notice that the JSONPath expression of the first object map of this triples map (i.e., line 12 in Listing 2) selects the JSON array that is the value of the "bbb" field of the JSON document, rather than selecting the elements within this array. Therefore, by applying our approach, the RDF term produced from this object map is a cdt:List literal with the string "[11,22]" as its lexical form (quotation marks not included). Similarly, the JSONPath expression of the second object map (line 15 in Listing 2) selects the whole JSON object that is the value of the "ccc" field, which, by our approach, results in the creation of a cdt:Map literal with a string such as "{"xxx": 43, "yyy": [3,4]}" as its lexical form.[3] Hence, the whole RDF graph produced for this example consists of two RDF triples as given in Listing 3.

Such RDF data can then be queried using SPARQL with the CDT extension introduced in our earlier work [6, 7]; this extension includes functions to operate on cdt:List and cdt:Map literals in expressions (as used in, e.g., FILTER and BIND clauses) and an UNFOLD operator to decompose the composite values represented by such literals into their individual components. As an illustrative example, consider the query in Listing 4 which, first, uses the cdt:size function to obtain the size of a list that is represented as a cdt:List literal and, then, uses this size to extract the last element of that list. Hence, for our running example with the RDF graph in Listing 3, the query returns an xsd:integer literal with value 22.[4]

---

[3]We emphasize that the exact lexical form of the produced cdt:List and cdt:Map literals depends on how the RML processor serializes JSON arrays and JSON objects (e.g., whether and where whitespace characters are included). Yet, as long as this serialization is a syntactically correct JSON representation of these arrays and objects, it is guaranteed that the resulting cdt:List and cdt:Map literals are well-formed and, thus, the values that these literals represent are unambiguously defined.

[4]As per the SPARQL-CDT spec [7], the individual elements within the lexical forms of cdt:List and cdt:Map literals are interpreted using the shorthand notation that the RDF Turtle syntax introduces for specific kinds RDF terms, which means that the substrings "11" and "22" within "[11,22]" are expanded to xsd:integer literals with values 11 and 22, respectively.

```
1  PREFIX cdt:   <http://w3id.org/awslabs/neptune/SPARQL-CDTs/>
2  PREFIX ex:    <http://example.org/>
3  SELECT ?lastElmt WHERE {  ?x  ex:test1  ?list
4                            BIND( cdt:size(?list) AS ?size )
5                            BIND( cdt:get(?list, ?size) AS ?lastElmt )   }
```

**Listing 4:** A SPARQL query to extract the last element of a list that is represented as a cdt:List literal and that may have been produced from a JSON array (as illustrated through Listings 1–3).

```
1  PREFIX cdt:   <http://w3id.org/awslabs/neptune/SPARQL-CDTs/>
2  PREFIX ex:    <http://example.org/>
3  SELECT ?k WHERE {  ?x  ex:test2  ?map
4                     BIND( cdt:keys(?map) AS ?keys )
5                     UNFOLD( ?keys AS ?k )     }
```

**Listing 5:** A SPARQL query to extract the field names (keys) of a map that is represented as a cdt:Map literal and that may have been produced from a JSON object (as illustrated through Listings 1–3).

As another example query, consider Listing 5. The cdt:keys function in this query can be applied to cdt:Map literals. For every such literal, the function returns a cdt:List literal that represents a list of all field names of the map represented by the given cdt:Map literal. The query then decomposes each such list using the UNFOLD operator. For the RDF graph in Listing 3, the query result consists of two solution mappings: one with the literal "xxx"^^xsd:string, the other with the literal "yyy"^^xsd:string.

We emphasize that the effect of the first of these two example queries can be achieved also for RDF data produced based on the Collections and Containers module of RML, but with a more complex query (because the size of an RDF collection can only be obtained via a subquery with a COUNT over the result of a property path pattern). In contrast, the effect of the second example query can *not* be achieved at all by using RML *without* our approach, because RML mappings do not have access to the field names in JSON files (due to the limited expressiveness of JSONPath).

We also notice, however, that the Collections and Containers module of RML provides more flexibility in terms of selecting which elements of the source data should be included in the created collection, while our approach is focused only on representing complete sub-parts of the JSON data as composite values. Yet, for use cases in which whole JSON arrays are meant to be translated into RDF collections, our approach is still an alternative to the Collections and Containers module of RML. This is the case because every cdt:List literal can be converted into such a collection and this conversion can be expressed as a SPARQL query (using the CONSTRUCT result form) [7, Section 3.4].

## 3. Behavior of Existing Implementations

In this section we briefly investigate whether our approach can already be used with existing RML processors. In particular, we are interested in how they handle rml:reference properties with JSONPath expressions that select JSON arrays or JSON objects rather than scalar values. To this end, we test the behavior of four such processors when presented with the example mapping of Listing 2, to be applied to the JSON file in Listing 1. The four[5] RML processors that we consider are BURP (version 0.1.1), RMLMapper (version 8.1.0), CARML (version 1.4.0-0.4.11), and Morph-KGC (version 2.10.0). Since CARML and Morph-KGC do not yet support the new RML vocabulary, we test them using an equivalent variation of the mapping in Listing 2 that is expressed in terms of the earlier version 1.1.2 of the RML vocabulary (which uses terms of the R2RML vocabulary). Table 1 summarizes the behavior.

For the JSONPath expression that selects a JSON array (line 12 in Listing 2), we observe that BURP produces the desired literal; that is, this literal is indeed a well-formed cdt:List literal that represents a list of the two integers from the JSON array at line 1 of Listing 1. In contrast, each of the other three RML processors produces two separate literals. Apparently, these processors misinterpret the JSONPath expression to select each of the elements within the JSON array rather than selecting the JSON array itself (the JSONPath expression to actually select the elements of the array at this point is bbb[*]).

---

[5]We also tried with SDM-RDFizer (version 4.7.5.13.5) but did not get it to work.

**Table 1**
Output of four RML processors for the object maps of Listing 2, when applied to the JSON file in Listing 1.

| System | Output for object map in line 12 | Output for object map in line 15 |
|---|---|---|
| BURP | one literal: | one literal: |
| (v. 0.1.1) | `"[11, 22]"^^cdt:List` | `"{xxx=43, yyy=[3, 4]}"^^cdt:Map` |
| RMLMapper | two literals: | one literal: |
| (v. 8.1.0) | `"11"^^cdt:List and "22"^^cdt:List` | `"{xxx=43, yyy=[3, 4]}"^^cdt:Map` |
| CARML | two literals: | one literal: |
| (v. 1.4.0-0.4.11) | `"11"^^cdt:List and "22"^^cdt:List` | `"{xxx=43, yyy=[3, 4]}"^^cdt:Map` |
| Morph-KGC | two literals: | nothing |
| (v. 2.10.0) | `"11"^^cdt:List and "22"^^cdt:List` | |

For the JSONPath expression that selects a JSON object (line 15 in Listing 2), we observe that none of the tested processors produces the desired literal. While Morph-KGC does not produce anything for this case, the other three processors produce at least the correct number of literals, namely one. Notably, the lexical form of these literals is identical, which might be attributed to the fact that the three processors are implemented in Java and use the same JSON library to represent JSON objects internally. However, this lexical form created for these literals is neither a valid JSON string (the field names are lacking their surrounding quotation marks and the =-characters should be colons instead) nor any other possible string in the lexical space of the cdt:Map datatype. Consequently, these literals are not well-formed cdt:Map literals and would not be recognized by the aforementioned CDT extensions of SPARQL; in other words, a function such as cdt:keys in Listing 5 would produce an error for these literals.

In summary, with one exception, our approach cannot be used yet with the tested RML processors, where the exception is the correct creation of cdt:List literals by BURP. The other tested processors may achieve the same correct behavior for cdt:List literals as BURP by fixing their interpretation of JSONPath expressions that select JSON arrays. For the creation of well-formed cdt:Map literals, BURP, RMLMapper, and CARML need to adapt their functionality for serializing JSON objects.

## 4. Demonstration of a Native Implementation of the Approach

To demonstrate the full approach at the workshop we have implemented the approach in the RML-processing component of our query federation engine HeFQUIN. In this section, we first provide an overview of this component and, thereafter, describe how the approach can be used in HeFQUIN.

The RML component of HeFQUIN is a dedicated implementation of RML that employs our recently-introduced algebra for declarative mapping languages [8]. That is, during startup, the component translates any given RML mapping into an expression of this algebra by applying the translation algorithm of our earlier work [8, Algorithm 1].[6] At runtime, such an algebra expression is translated into a physical mapping plan that, then, is executed to convert the provided source data into RDF. For the time being, the only type of source data that is supported is JSON data (which HeFQUIN retrieves from Web APIs as we shall discuss below). For the evaluation of JSONPath expressions during this process, HeFQUIN employs the Jayway JsonPath library[7] in a way that it correctly implements the approach introduced in this paper; i.e., for the example of Section 2, it produces the RDF graph in Listing 3.

By being a query engine, HeFQUIN is not designed to produce RDF representations of source data as its main purpose. Instead, it produces such RDF views internally for data obtained via requests to non-RDF data sources (in particular, JSON-based Web APIs), for the purpose of evaluating sub-patterns of a given SPARQL query over these views. Consequently, the effect of applying RML mappings in this context can be observed through the query results.

To enable users to express that some part of a SPARQL query is meant to be evaluated over an RML-based RDF view of data from a Web API, HeFQUIN supports an extended notion of SERVICE clauses. The extension consists of the option to include a so-called PARAMS clause between the service IRI and the graph pattern of the SERVICE clause, as illustrated in the example query in Listing 6. Such

---

[6]In fact, while the translation algorithm in our earlier work was still designed for version 1.1.2 of the RML vocabulary, for HeFQUIN we have adapted this algorithm to the new RML vocabulary [2].

[7]https://github.com/json-path/JsonPath

```
1   PREFIX xsd:   <http://www.w3.org/2001/XMLSchema#>
2   PREFIX cdt:   <http://w3id.org/awslabs/neptune/SPARQL-CDTs/>
3   PREFIX ex:    <http://example.org/>
4   SELECT * WHERE {
5      BIND ( "52.52"^^xsd:float AS ?lat )   BIND ( "13.41"^^xsd:float AS ?long )
6      SERVICE <http://example.org/OpenMeteoWrapper> PARAMS( ?lat AS "latitude",
7                                                            ?long AS "longitude" ) {
8         ?x ex:hourlyTemperature ?list .
9      }
10     BIND ( (cdt:get(?list, cdt:size(?list)) - cdt:get(?list, 1)) AS ?tempDiff )
11  }
```

**Listing 6:** A SPARQL query that calculates the difference between the first and the last temperature of an hourly temperature forecast retrieved as a JSON array from a Web API and mapped to a cdt:List literal.

```
1   { "latitude" : 52.52,
2        ...
3      "hourly" : { "time": ["2026-03-08T00:00","2026-03-08T01:00","2026-03-08T02:00", ... ],
4                   "temperature_2m": [7.1, 6.7, 6.2, 5.9, 5.8, 5.4, 5.2, 5.8, ... ] }
5   }
```

**Listing 7:** Example of JSON data returned by the Open-Meteo API.

a PARAMS clause can be used to identify the query variables that provide values for parameters used in a URI template for creating the request URIs for the particular API. A so-called federation description defines these URI templates for all Web APIs to be used as members of the federation considered by HeFQUIN. As the exact details of the URI templates and their instantiation through a PARAMS clause are not relevant for our approach in this paper, we refer to the corresponding part of the HeFQUIN documentation for these details.[8] The following request URI would be formed for the query in Listing 6.

https://api.open-meteo.com/v1/forecast?hourly=temperature_2m&latitude=52.52&longitude=13.41

This URI accesses the Open-Meteo API[9] to retrieve a JSON document with a live temperature forecast for the provided geographic coordinates. Listing 7 illustrates a relevant snippet of this JSON data.

When executing the query in Listing 6, HeFQUIN creates an RDF view of this JSON data and evaluates the graph pattern of the SERVICE clause over this view, where the view is defined based on an RML mapping that is part of the federation description provided to HeFQUIN. Listing 8 presents a fragment of such a federation description that introduces the Open-Meteo API as a federation member to be considered by HeFQUIN, including a simple example of an RML mapping to be used for the JSON documents retrieved via this API. We emphasize that the object map of this mapping (i.e., lines 20–21 of Listing 8) references a JSON array within the JSON data and, as per our approach, maps the selected array into a cdt:List literal. The example query can then operate on this literal (see line 10 in Listing 6).

While the example in this section focuses on a query with a single API call, HeFQUIN supports queries that result in multiple calls to the same API, and also to different APIs. In other words, by writing queries that contain multiple SERVICE clauses, HeFQUIN can be used to join data from multiple data sources, which may not only be Web APIs but also RDF-based data sources (e.g., SPARQL endpoints, TPF servers). For instance, it is possible to extend the query in Listing 6 such that the geographic coordinates of relevant locations are retrieved from, say, the DBpedia knowledge graph. For the demonstration at the workshop we have prepared a number of example queries, which can already be tried now.[10]

## Acknowledgments

The work presented in this paper was supported by the Knut and Alice Wallenberg Foundation (KAW 2023.0111) and by the Swedish Research Council (Vetenskapsrådet, project reg. no. 2025-06246).

---

[8] https://liusemweb.github.io/HeFQUIN/doc/queries.html#service-calls-with-parameters
[9] https://open-meteo.com/
[10] https://liusemweb.github.io/HeFQUIN/doc/examples.html

```
1   PREFIX xsd:     <http://www.w3.org/2001/XMLSchema#>
2   PREFIX fd:      <http://w3id.org/hefquin/feddesc#>
3   PREFIX hydra:   <http://www.w3.org/ns/hydra/core#>
4   PREFIX rml:     <http://w3id.org/rml/>
5   PREFIX ex:      <http://example.org/>
6
7   ex:openMeteo fd:serviceURI <http://example.org/OpenMeteoWrapper> ;
8                fd:interface  _:openMeteoRestEndpoint ;
9                fd:wrapper    [ fd:rmlTriplesMaps ( _:openMeteoTM1 ) ] .
10
11  _:openMeteoRestEndpoint a fd:RESTInterface ;
12      fd:iriTemplate [ # ... description of the template for creating the request URIs
13                     ] .
14
15  _:openMeteoTM1 rml:logicalSource [ rml:source _:openMeteoRestEndpoint ;
16                                     rml:referenceFormulation rml:JSONPath ;
17                                     rml:iterator "$.hourly" ] ;
18      rml:subjectMap [ rml:termType rml:BlankNode ] ;
19      rml:predicateObjectMap [ rml:predicateMap [ rml:constant ex:hourlyTemperature ] ;
20                               rml:objectMap [ rml:reference "temperature_2m" ;
21                                               rml:datatype cdt:List ] ] .
```

**Listing 8:** A snippet of an RDF-based federation description (presented in the RDF Turtle syntax) that introduces the Web API of the Open-Meteo weather service as a federation member to be considered by our approach.

## Declaration on Generative AI

The author has not employed any Generative AI tools.

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
