# OpenReview forum: "The HeFQUIN Approach to Extract and to Process Composite Values with RML and SPARQL"
_eswc-conferences.org/ESWC/2026/Workshop/KGCW — KGCW 2026_

### Official Review · ~Enayat_Rajabi2 · 2026-03-13
**Demo paper**

**Rating:** 7
**Confidence:** 5

**Review:**

The paper is going to extend the RDF Mapping Language so it can generate composite data structures (collections and containers) when mapping heterogeneous data sources into RDF knowledge graphs. The author extends this feature in the federation query engine HeFQUIN. It maps JSON arrays and objects directly into CDTs—specifically cdt:List and cdt:Map. The paper addresses a pain point in the RDF community and simplifies the SPARQL query logic. The use of cdt:keys combined with the UNFOLD operator represents a functional improvement over existing knowledge graph construction methods.

I think it is a good idea to have such a paper as a demo paper for the KGC workshop. It would be interesting to see how the approach has been implemented within their HeFQUIN federation engine—specifically the use of SERVICE clauses with PARAMS to fetch and map live data from a API.

Some recommendations for improvements:
- A brief footnote or table clarifying the exact mapping differences between RML 1.1.2 and the new modular RML would be helpful for readers.
- It is heavily focused on JSON-based sources. While the author acknowledges that extending this to other formats (e.g., CSV, XML) is future work, it would be good to mention a sentence on the potential challenges of "casting and serialization" for non-nested formats would strengthen the "Approach" section.

The paper provides a technically sound, well-motivated, and an extension to RML that leverages existing work on CDTs to solve real-world data modelling challenges. The topic is also relevant to the KGC community.

Minor comments:  “if an an object map” —> “if an object map”

---

### Official Review · ~Davan_Chiem_Dao1 · 2026-03-17
**Review of "The HeFQUIN Approach to Extract and to Process Composite Values with RML and SPARQL"**

**Rating:** 8
**Confidence:** 4

**Review:**

## Summary

Previous work introduced composite datatypes (CDTs), RDF literals designed to represent composite values such as lists and maps, together with SPARQL-CDT functions and operators such as cdt:size, cdt:get, cdt:keys, and UNFOLD for querying such literals. This demonstration paper presents an extension of the HeFQUIN SPARQL processor to support these CDTs in practice. More specifically, it shows how JSON arrays and objects can be mapped through RML into cdt:List and cdt:Map literals, which can then be queried with SPARQL-CDT. The paper highlights that this approach provides a simpler way to query lists than the current RDF collection standard, and also enables querying map-like values in a way that is not directly supported by standard RDF representations.

## Strengths

- The paper is well written and easy to follow.
- Resources to test the engine is provided and easy to follow.
- The proposal is compared and contrasted with existing approaches, mainly RML:CC.
- It provides the first concrete implementation of the CDT idea, extending previous work that introduced the concept.
- The contribution is not limited to SPARQL querying, but also covers generation through RML, which makes the proposal more practical and complete.
- The approach offers a simpler alternative to querying lists compared with standard RDF collections.
- It also introduces support for map-like composite values, which are not naturally handled in standard RDF.

## Limitations and Recommendations

- The scope is currently limited to JSON sources, where the mapping to composite datatypes is very close to the source structure. As a consequence, the mapping seems somewhat trivial.
- The paper doesn't discuss in much depth how the approach would extend to other source formats such as XML, where the mapping may be less straightforward.
- It would be useful to justify why CDTs should be preferred over alternatives such as rdf:JSON.

## Overall Assessment
Overall, this is a clear and interesting demonstration paper. Its main contribution is to make the CDT idea concrete through an implementation in HeFQUIN and by connecting RDF generation with SPARQL querying. While the scope is still somewhat narrow, extending to other types of sources is already planned as future work by the author.

### Typos

Section 2 Paragraph 1:
"if an an object map" -> "if an object amp"

---

### Official Review · ~Mario_Scrocca1 · 2026-04-02
**Valuable demo paper on CDTs support in RML and HeFQUIN tool features**

**Rating:** 8
**Confidence:** 5

**Review:**

This demo paper introduces a practical approach to generating composite RDF literals (lists and maps CDTs) directly from JSON data within RML mappings. The solution is implemented in the HeFQUIN query federation engine and demonstrates how SPARQL-CDT functions can operate on these composite values for real Web API data, virtualised to RDF via RML mappings.

The demonstration is definitely relevant to the workshop, and support for CDTs can spark interesting discussions related to the long‑standing challenge of how RML mapping engines should handle multi‑value references in nested data structures. I like that the paper highlights the distinction between references targeting an entire array versus mappings targeting its elements (array[*]). Indeed, many processors do not differentiate the behaviour between the two cases (where the behaviour depends on the adopted JSON library), whereas the new spec includes a test case requiring explicit throwing of an error in these cases (cf. https://kg-construct.github.io/rml-core/test-cases/docs/#RMLTC0025b-JSON).

The paper is well written, easy to follow, and supported by good examples that clearly illustrate the proposed approach. I particularly appreciate that the author included an empirical evaluation across multiple existing RML processors, which provides a realistic picture of current behaviours and highlights inconsistencies that this work helps clarify.

Apart from the specific case considered for CDTs mapping via RML, I believe the demonstration of the HeFQUIN tool and the considered example applying RML mappings to query REST API(s) can be also very interesting for the workshop participants. Relevant material is linked in the paper for interested readers.

One limitation of the paper is the lack of discussion on extending the approach beyond JSON to other common formats such as CSV and XML. While the paper briefly mentions that this is part of future work, elaborating on what such extensions would require (e.g., serialization rules, datatype casting, handling of nested structures) would make the contribution more complete and highlight its generalizability.

A minor comment concerns footnote 5: it is unclear whether SDM‑RDFizer could not be executed in the author’s environment or whether the tool inherently fails for the considered scenario.

---

### Decision · Program_Chairs · 2026-04-09

**Decision:**

Accept

**Comment:**

This paper has been selected for presentation at the KGC workshop. We strongly encourage the authors to consider the reviews whilst revising the paper. Camera-ready instructions will soon follow.